# Effects of Yeast on the Growth and Development of *Drosophila melanogaster* and *Pardosa pseudoannulata* (Araneae: Lycsidae) Through the Food Chain

**DOI:** 10.3390/insects16080795

**Published:** 2025-07-31

**Authors:** Yaqi Peng, Rui Liu, Wei Li, Yao Zhao, Yu Peng

**Affiliations:** 1Hubei Key Laboratory of Regional Development and Environmental Response, Faculty of Resources and Environmental Science, Hubei University, Wuhan 430062, China; 201911110711069@stu.hubu.edu.cn (Y.P.); liwei@hubu.edu.cn (W.L.); 2Hubei Insect Resources Utilization and Sustainable Pest Management Key Laboratory, College of Plant Science and Technology, Huazhong Agricultural University, Wuhan 430070, China; apk14@outlook.com; 3State Key Laboratory of Biocatalysis and Enzyme Engineering, School of Life Sciences, Hubei University, Wuhan 430062, China

**Keywords:** yeast, *Drosophila melanogaster*, *Pardosa pseudoannulata*, growth and development, nutrients

## Abstract

*Pardosa pseudoannulata* is a natural predator that helps control insect pests. Since yeast is known to improve the growth and reproduction of *Drosophila melanogaster*, we tested whether yeast in the flies’ diet affects both the flies and the spiders that eat them. We compared three diets for the flies: no yeast, active yeast, or inactivated yeast. Adding yeast helped fruit flies grow faster, increased their body weight, and boosted their fat, protein, and glucose levels. When spiders ate these yeast-fed flies, their early development (second instar) was faster, but yeast had no effect on their final size or weight. Yeast also increased the spiders’ fat content but did not change their protein or glucose levels. Our findings highlight how yeast in fruit-fly diets influences both flies and their spider predators.

## 1. Introduction

Spiders are predators and the most prominent generalist predators in rice fields [1]. They play a key role in controlling rice-field pests [2,3], Although spider nutrition has been studied by many researchers, the effect of the presence of yeast in fruit-fly culture on spiders through the food chain has been poorly studied [4,5,6,7,8]. Studying the nutritional ecology of spiders is critical because it can aid in understanding the evolution of prey capture and life history strategies, factors that regulate both abundance and diversity [9,10]. The nutrient composition of the prey’s diet affects the growth and survivorship of a generalist predator [11,12,13]. Food resources are essential for all aspects of an animal’s life, and the amount and types of food consumed can have an enormous influence on growth, reproduction, and survival [14,15,16,17]. *Pardosa pseudoannulata* preys mainly on planthoppers, leaf rollers, and leaf hoppers in rice fields [18]. Based on the characteristics of the biology, behaviour, and ecology of *P. pseudoannulata*, it has been concluded that this spider has a long growth and reproduction period, many egg laying periods, a large reproductive output, and high hatching and survival rates, and it is suitable for artificial breeding and has a high protection and utilisation value [3,10]. Food is essential to an animal’s behaviour and life history, including growth, reproduction, and survival [15,19]. The impact of spiders on natural and agro-ecosystems depends on the number of spiders as well as their habitat selection and foraging activities [20,21,22]. Therefore, it is crucial to understand the effects of factors such as prey nutrition on spider survival, foraging, growth, mating, and reproduction. It has been shown that the quality of prey affects the growth and reproduction of predatory arthropods, but little is known about the specific nutrients responsible for these effects [23,24]. Research on the nutritional ecology of spiders has mainly focused on how macronutrients such as protein and lipids affect the life history characteristics of spiders [9,25,26]. There are also some influences of trace elements. For instance, an important determinant of nutritional quality is the content of polyunsaturated fatty acids (PUFAs) [27]. The nutritional content of spiders themselves is directly related to the nutritional content of the food they prey upon [28,29]. Yeast contains a lot of nutrients, such as vitamins, proteins, fats, etc., which are needed for the survival of adult fruit flies and also promote the growth and development of their larvae.

The inclusion of yeast in the medium used for fruit-fly culture can promote the growth and reproduction of *D. melanogaster* [30,31]. The survival rate of *D. melanogaster* larvae is low when there is no yeast or low yeast intake. Increasing the ratio of protein to carbohydrate in the larval food can improve survival, shorten the larval development time, and increase adult quality [32,33,34,35]. Protein and carbohydrate are the two major macronutrients that exert profound influences over fitness in many organisms, including *Drosophila melanogaster* [36]. Studies have shown that altering the content of macronutrients in the medium in which *D. melanogaster* larvae develop results in changes in the macronutrient content in adult *D. melanogaster*, and these changes affect the survival, growth, and reproduction of spiders that feed on *D. melanogaster* [26,37,38,39,40,41,42]. In this paper, three treatments (no yeast, active yeast, and inactivated yeast) were applied to the conventional medium formulated for *D. melanogaster*. The effects of different yeast treatments on the pupation and emergence of *D. melanogaster* larvae, emergence rate, development time, adult body weight, and nutritional composition were analysed. The growth and development of *P. pseudoannulata* were also assessed after feeding on *D. melanogaster* grown on different media. We hypothesised that the presence of yeast in fruit-fly culture could improve the quality of the *P. pseudoannulata*. In order to rule out other influences, we only used *D. melanogaster* for feeding.

## 2. Materials and Methods

### 2.1. Collection and Rearing

We collected the *Pardosa pseudoannulata* in the rice field at Huazhong Agricultural University, Wuhan, Hubei Province, China, between September and December 2019 [18]. The collected *P. pseudoannulata* was reared at a temperature of 25 ± 1 °C, relative humidity of 60–70%, and light conditions of 14L:10D. A wet sponge was placed at the bottom of a glass finger tube for feeding, and the mouth of the tube was plugged with cotton. The spider was fed every 4 days with 10 fruit flies. The hatchlings underwent their first moult within the egg sacs. Only juveniles from the first clutch were assigned to the prey nutrient treatment groups (see below).

### 2.2. Generation of Fruit Flies with Different Nutrients

To generate fruit flies with different nutrients as prey for *P. pseudoannulata* juveniles, we prepared three types of culture medium. Group AY culture medium (activated yeast), the standard fruit-fly culture medium, comprised 240 mL H_2_O, 22 g corn powder, 16 g sucrose, 4 g yeast (Angel, Yichang, China), 1.6 g agar (Phygene, Fuzhou, China), 0.1 g benzoic acid (Sinopharm, Shanghai, China) (dissolved in 2 mL ethyl alcohol), and 1 mL propanoic acid (Sinopharm, Shanghai, China) [31]. Group IY culture medium contained 4 g of inactivated yeast. Group NY culture medium did not contain yeast. All other components of Groups IY and NY were identical to those of Group AY culture medium.

### 2.3. Juvenile Survival and Growth

Eight replicates per group were used for each of the three media. After the media were cooled and solidified, each medium was inoculated with five pairs of *D. melanogaster* within 8 h of fledging. *D. melanogaster* was transferred to a new glass tube 6 h after inoculation, and the larval duration, pupal duration, developmental duration, the number of pupae, and the number of flies emerged were observed and recorded each day. Forty newly emerged (unmated) *D. melanogaster* (female: male = 1:1) were randomly selected from three different media and frozen at −20 °C, and then each *D. melanogaster* was weighed using an analytical balance (Mettler Toledo, Greifensee, Switzerland). Since male and female *D. melanogaster* individuals are inherently different in size, we compared the effects of different treatments on female and male individuals separately.

The spiderlings (second instar) hatched from the egg sacs were divided into three groups, and each group consisted of 60 individuals. The juvenile spiders were housed individually in glass tubes (diameter × length: 20 × 60 mm) that were plugged with absorbent cotton. The spider was fed every 4 days with 10 fruit flies. The hatchlings underwent their first moult within the egg sacs. Only juveniles from the first clutch were assigned to the prey nutrient treatment groups (see below). The NY group was fed *D. melanogaster* that had been reared without yeast. The AY group was fed *D. melanogaster* that had been reared on medium containing activated yeast. The IY group was fed *D. melanogaster* that had been reared on medium containing high-temperature-inactivated yeast. In order to avoid the influence of other factors, only *D. melanogaster* was provided as food during the experiment. The spiderling was fed three times a week: five *D. melanogaster* each time. We replenished the water and cleaned any residue from the tube on each occasion. The moulting time and survival of *P. pseudoannulata* were recorded until they died. Twenty fifth-instar *P. pseudoannulata* in each of the three treatments were randomly selected. After being anaesthetised with CO_2_ gas, body length was measured under an optical microscope (SZX7), and they were weighed using a precision balance (BT1251).

### 2.4. Total Fat, Protein, and Glucose Were Measured

Five newly emerged *D. melanogaster* were randomly selected from the three different media and placed in a 1.5 mL centrifuge tube. We firstly added 50 μL 1 × PBS buffer into the test tube, and after the D. melanogaster was fully ground with three small steel balls, we used the buffer to rinse these steel balls into the test tube twice. Then, we shook the test tube violently with a turbine mixer for 1 min. Finally, we centrifuged the mixture in the test tube with a high-speed centrifuge at 4 °C, 10,000 r/min, for 10 min. A total of 200 μL of supernatant was taken for subsequent use, and each treatment was repeated six times. The content of protein was determined using a Protein Quantification Kit (Bradford Assay). The content of glucose was determined using the CheKineTM Glucose Assay Kit. The total fat content was determined by the acid hydrolysis method.

Ten spiderlings of the fifth instar were selected from the three treatment groups and stored in a −20 °C refrigerator, and the body substance content of each young spider was determined. The measurement method of nutrients in *P. pseudoannulata* was the same as for *D. melanogaster*.

### 2.5. Statistics

The difference in total fat, protein, and glucose content in *D. melanogaster* and *P. pseudoannulata* in different media was compared by one-way ANOVA. All values are reported as mean ± standard deviation (mean ± SD), unless otherwise stated. The developmental period, body length, and body weight of *P. pseudoannulata* were compared by one-way ANOVA. The difference in survival rate was measured by log-rank [43]. Statistical analysis of the data was performed using Excel and SPSS version 19 (IBM Corporation, Armonk, NY, USA), and GraphPad Prism 7 software was used for figure drawing.

## 3. Results

### 3.1. Growth and Development of Drosophila melanogaster

When comparing *D. melanogaster* fed with three different yeast treatments, significant differences were observed in the larval period (Table 1; one-way ANOVA: *F*_2,33_ = 14.933, *p* < 0.001), pupal period (*F*_2,33_ = 23.238, *p* < 0.001), total developmental time (*F*_2,33_ = 50.667, *p* < 0.001), the number of pupations (*F*_2,33_ = 235.823, *p* < 0.001), the number of eclosions (*F*_2,33_ = 158.205, *p* < 0.001), and the eclosion rate (*F*_2,33_ = 6.956, *p* = 0.01). Specifically, post hoc paired comparisons revealed that the pupal period (independent *t*-test: *t* = −3.024, *df* = 8, *p* = 0.016) and total developmental time of the AY group of *D. melanogaster* were significantly shorter than those of the IY group (*t* = −4.382, *df* = 8, *p* = 0.002). The larval period (*p* = 0.004), pupal period (*p* = 0.003), and total developmental time of the IY group of *D. melanogaster* were significantly shorter than those of the NY group (*p* < 0.001), while the larval period of the AY group did not differ significantly from that of the IY group (*p* = 0.215). The pupal period and total developmental time of *D. melanogaster* on activated yeast were significantly shorter than on inactive yeast. The number of pupations and eclosions of the AY group of *D. melanogaster* was significantly higher than that of the IY group (*p* < 0.001), while the eclosion rate did not differ significantly between these two groups (*p* = 0.616). The number of pupations (*p* < 0.001) and the number of eclosions (*p* < 0.001), as well as the eclosion rate of the IY group of *D. melanogaster*, were significantly higher than those of the NY group (*p* = 0.012).

Comparison of the weights of different sexes within the three groups of *D. melanogaster* showed significant differences (Figure 1; female: *F*_2,166_ = 31.463, *p* < 0.001; male: *F*_2,150_ = 28.556, *p* < 0.001). The weights of both male and female *D. melanogaster* in the AY group were significantly higher than those in the IY group (*p* < 0.001), while there was no significant difference in the weights of male *D. melanogaster* between the IY and NY groups (*p* = 0.231). However, there was a significant difference in the weights of female *D. melanogaster* (*p* = 0.01).

### 3.2. Survival and Growth of Spiderlings

The survival of juvenile *P. pseudoannulata* differed significantly according to the diet of the *D. melanogaster* on which they were reared (Figure 2; log-rank test: *χ*^2^= 18.23, *df* = 2, *p* < 0.001). The survival rate of juvenile spiders fed on the AY group of *D. melanogaster* was significantly lower than that of those fed on the IY group of *D. melanogaster* (*χ*^2^ = 4.481, *p* = 0.0343), and the survival rate of those fed on the IY group of *D. melanogaster* was significantly lower than that of those fed on the NY group of *D. melanogaster* (*χ*^2^ = 30.96, *p* < 0.001).

Overall, the prey type significantly affected the duration of the second instar of the spiderlings (Table 2; *F*_2,36_ = 5.330, *p* = 0.009) but had no significant impact on the developmental times of the third (*F*_2,36_ = 2.127, *p* = 0.134), fourth (*F*_2,36_ = 1.068, *p* = 0.354), and fifth instars (*F*_2,36_ = 2.000, *p* = 0.150). The developmental time of spiderlings feeding on AY fruit flies in the second instar was significantly lower than those feeding on IY fruit flies (*p* = 0.01), with no significant difference between the IY group and the NY group. Similarly, there were no significant differences in the body length (Figure 3a,b; *F*_2,27_ = 0.129, *p* = 0.879) or weight of the spiders between these two treatments (*F*_2,27_ = 0.030, *p* = 0.970).

### 3.3. The Content of Nutrients in D. melanogaster and P. pseudoannulata

There were significant differences in protein content (*F*_2,32_ = 20.076, *p* < 0.001), glucose content (*F*_2,32_ = 31.027, *p* < 0.001), and total fat content (*F*_2,15_ = 20.719, *p* < 0.001) (Figure 4) in *D. melanogaster* fed with the three different media. There were no significant differences in total fat (*p* = 0.196) or protein contents between the AY and IY groups (*p* = 0.086), but the IY group was significantly higher than the NY group (*p* < 0.001). The glucose content of *D. melanogaster* in the AY group was significantly higher than that in the IY group (*p* < 0.001), and that in the IY group was significantly higher than that in the NY group (*p* = 0.002).

The three different culture media-fed *D. melanogaster* had no significant impact on the glucose (*F*_2,15_ = 1.272, *p* = 0.309) or protein content in the *P. pseudoannulata* (*F*_2,15_ = 0.931, *df* = *2*, *p* = 0.416) (Figure 5). However, there was a significant difference in total fat content (*F*_2,15_ = 81.521, *p <* 0.001). The total fat content in the *P. pseudoannulata* of the AY group was significantly higher than that of the IY group (*p <* 0.001), and the IY group was significantly higher than the NY group (*p <* 0.001).

## 4. Discussion

Research on the impact of food on the development of *D. melanogaster* usually focuses on the types and choices of food during the larval to adult stages or only during the adult stage, with little research involving the larval developmental period [41]. Studies have shown that the quantity and quality of yeast in larval food can affect the early development and adult life traits of *D. melanogaster* [44]. The body weight and length of *D. melanogaster* are important indicators of whether they are developing well [45]. In our experiment, we found that adding active yeast to the medium can shorten the larval period, pupal period, and total development time of *D. melanogaster*, as well as increase the number of pupations and eclosions and the eclosion rate of the flies, consistent with previously reported results [46,47]. The number of pupae in the group fed active yeast was significantly higher than in the other two groups, suggesting that *D. melanogaster* lay more eggs on medium with active yeast. Adding yeast to the medium can significantly increase the fat, protein, and glucose content of *D. melanogaster*. At the same time, it can increase the number of eggs laid by *D. melanogaster*. The number of days taken by eggs to develop into pupae is greatly reduced, indicating that yeast powder promotes the growth and development of *D. melanogaster* larvae [32]. Medium without added yeast gradually becomes dry over time, making it difficult for first-instar larvae to move and ultimately leading to developmental stagnation [48]. In the current study, the number of individuals in the F1 generation of *D. melanogaster* reared on the medium without added yeast was significantly lower than that of the other groups [49]. Related studies have shown that in the absence of yeast, larval survival rates are very low, but that reducing the yeast content in larval food can significantly prolong the lifespan of adults [50].

Food is crucial for the growth, reproduction, and survival of animals [51]. Although many studies have shown that the quality of prey can affect the growth and reproduction of predators [23], there has been little research on the effects of yeast on spiders via fruit-fly culture medium. The nutritional components of the prey can affect a wide range of life-history traits in spiders. In this study, the effects of nutrient content in fruit-fly medium on the growth and development of *P. pseudoannulata* were studied. Our study fed spiders with fruit flies reared on different yeast treatments, and the results showed that fruit flies fed with activated yeast shortened the developmental time of the second instar of *P. pseudoannulata* spiders, with no significant effects on other instars. *P. pseudoannulata* spiders that were fed fruit flies reared without yeast had the longest survival time. This result may be due to the smaller size and nutritional deficiencies of the flies in the no-yeast group. When the three treatment groups were fed the same number of flies, the spiders in the no-yeast group took in a relatively small amount of food. Numerous studies have shown that restricting food, in terms of both caloric and nutrient content, extends the life span of most animals to some extent, including yeast, nematodes, spiders, fruit flies, mice, rats, and rhesus monkeys [52,53,54]. It is possible to induce dietary restriction in captive female *Frontinella pyramitela* spiders by varying the number of *Drosophila* flies given to individual spiders (five, three, or one flies/week). DR increased mean longevity (respectively, 42, 63, and 81 days) [55].

Our experiments have also shown that food restriction can increase the lifespan of spiders. In this study, only fruit flies were fed to the spiders to exclude the influence of other factors. The survival and growth rates of spiders feeding exclusively on *D. melanogaster* significantly decrease [56,57]. A previous study of *Hylyphantes graminicola* revealed that a diet of high-lipid prey was associated with increased mortality during development [29]. Other studies on spiders have also shown that although a single food source initially results in high survival, growth, and developmental rates, spiders feeding exclusively on fruit flies cannot develop to reproductive maturity [58,59], suggesting that fruit flies may not be an adequate food source for mature spiders. For example, research has shown that when female Linyphiidae spiders are fed only a large number of fruit flies, their egg quality sharply decreases [9,26,60]. Mixed food sources may benefit predators, as they provide a wider range of nutrients [61,62]. Because the subjects of this study were fifth instar *P. pseudoannulata* spiders and their body size was small, the size difference among the three treatments was not significant. The growth rate and age length of second instar spiders respond more significantly to environmental changes (such as temperature and food) than those of spiderlings [63]. Other studies have reported significant differences in spider development with mixed food sources, while differences in those with single food sources were smaller [31].

Animals need to supplement various nutrient components to meet their nutritional needs, and their food-seeking behaviour is also influenced by the nutritional composition of the food. Knowledge of animal nutritional requirements can help in understanding their physiology, behaviour, ecology, and evolution [64,65]. Our experimental results showed that adding yeast to the culture medium could significantly increase the fat, protein, and glucose contents in *D. melanogaster*, while there was no difference in the protein and glucose contents in *P. pseudoannulata* as a predator, and the total fat content in *D. melanogaster* was only significantly increased when fed with yeast. Spiders can extract almost all lipids from their prey, but the protein absorption rate is low [66].

In summary, our study found that yeast promoted the growth and development of *D. melanogaster* larvae but shortened the developmental time of the second instar of *P. pseudoannulata*. The contents of glucose, protein, and total fat in *D. melanogaster* were significantly increased by adding yeast to the *D. melanogaster* culture medium, but only the contents of total fat in *P. pseudoannulata.* Our study illustrates the importance of yeast to the growth and development of fruit flies. Since a single food cannot provide sufficient nutrition for spiders, we will conduct mixed feeding until spiders mature in the follow-up experiment, and then study the effects of yeast on spiders through the food chain.

## Figures and Tables

**Figure 1 insects-16-00795-f001:**
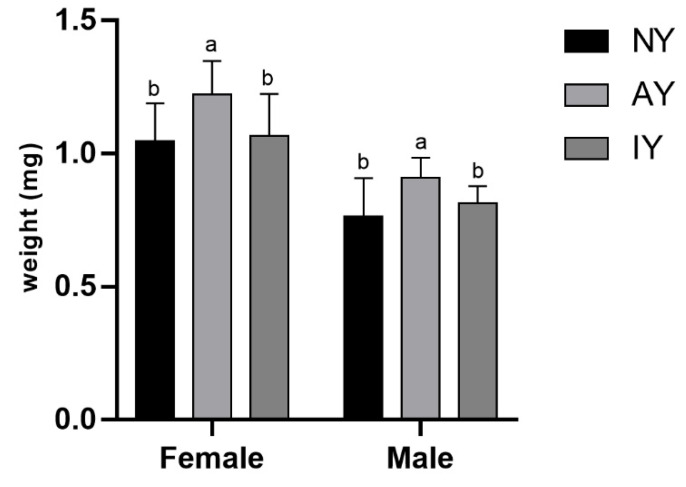
Body weight of *Drosophila melanogaster* reared on different yeast treatments. Different letters indicate that there are significant differences between treatments (*p* < 0.05).

**Figure 2 insects-16-00795-f002:**
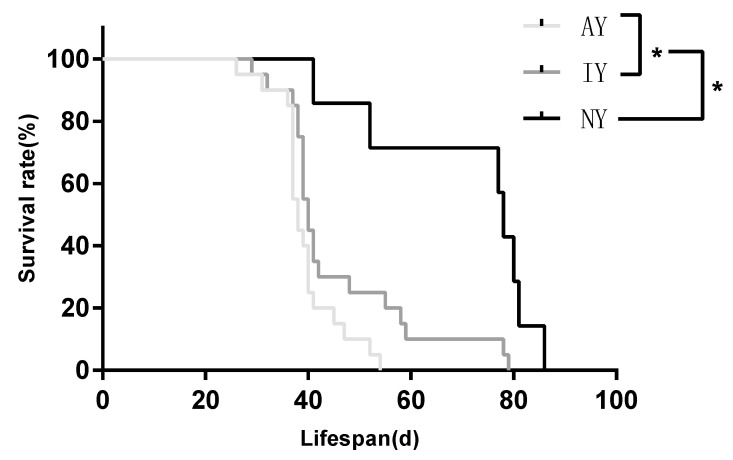
Survival rate and lifespan of *Pardosa pseudoannulata* fed *Drosophila melanogaster* reared on culture media with different yeast treatments (log-rank test, * *p* < 0.05).

**Figure 3 insects-16-00795-f003:**
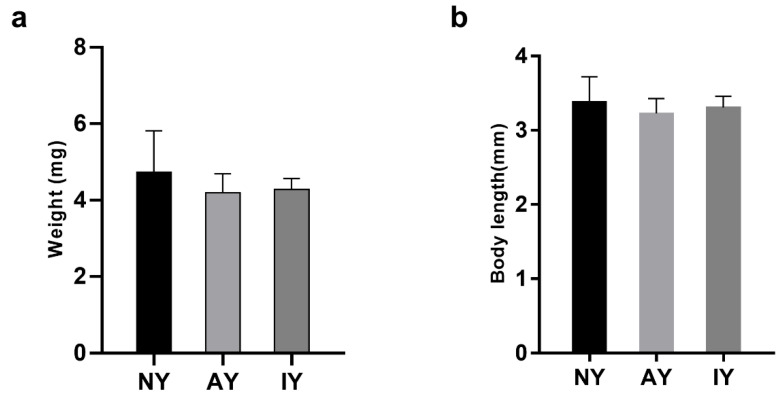
The body weight and length of *Pardosa pseudoannulata* fed *Drosophila melanogaster* reared on different yeast treatments. (**a**) The body weights of different treatment groups of *Pardosa pseudoannulata*; (**b**) The body lengths of different treatment groups of *Pardosa pseudoannulata*.

**Figure 4 insects-16-00795-f004:**
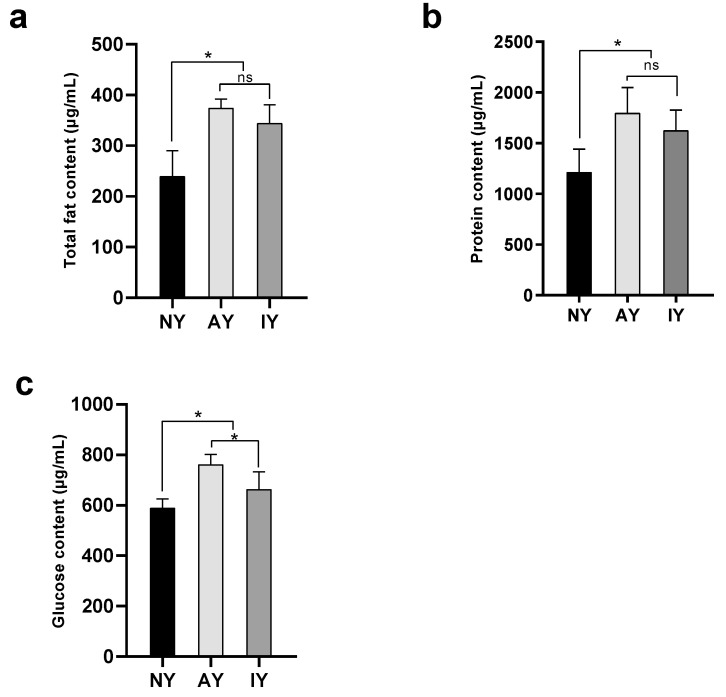
Total fat content (**a**), protein content (**b**), and glucose content (**c**) in *Drosophila melanogaster* treated with different yeasts. An asterisk indicates a highly significant difference between groups (LSD test, *p* < 0.05). The ns indicates no significant difference (LSD test, *p* > 0.05).

**Figure 5 insects-16-00795-f005:**
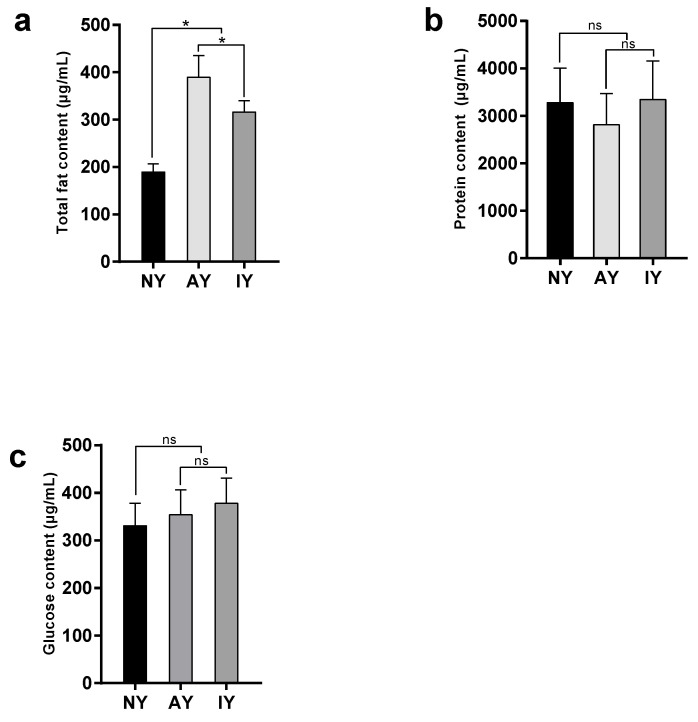
Total fat content (**a**), protein content (**b**), and glucose content (**c**) of *Pardosa pseudoannulata* reared by *Drosophila melanogaster* from the culture media with different yeast treatments. An asterisk indicates a highly significant difference between groups (LSD test, *p* < 0.05). The ns indicates no significant difference (LSD test, *p* > 0.05).

**Table 1 insects-16-00795-t001:** Biological properties of *D. melanogaster* reared on different yeast treatments.

	Treatment Group		
Biological Properties	NY	AY	IY
Larval duration (d)	7.4 ± 0.55 a	5.0 ± 0.71 b	5.8 ± 0.84 b
Pupal duration (d)	6.8 ± 0.84 a	3.2 ± 0.84 c	4.8 ± 0.84 b
Developmental duration (d)	14.2 ± 1.10 a	8.2 ± 0.84 c	10.6 ± 0.89 b
The number of pupations	62.8 ± 3.70 c	118.4 ± 5.86 a	99.4 ± 1.67 b
The number of eclosions	49.6 ± 3.29 c	100.4 ± 6.84 a	83.40 ± 2.41 b
Eclosion rate (%)	79.03 ± 0.04 b	84.74 ± 0.02 a	83.89 ± 0.01 a

The data are presented as mean ± SE. Different letters denote a significant difference (LSD test, *p* < 0.05).

**Table 2 insects-16-00795-t002:** Developmental duration of *Pardosa pseudoannulata* fed *Drosophila melanogaster* reared on culture media with different yeast treatments (d).

	Treatment Group		
Instar	NY	AY	IY
2nd instar	14.31 ± 2.78 a	11.15 ± 1.50 b	14.08 ± 3.50 a
3rd instar	7.69 ± 0.95 a	8.61 ± 1.04 a	8.31 ± 1.44 a
4th instar	7.69 ± 0.75 a	7.38 ± 0.51 a	7.77 ± 0.83 a
5th instar	11.08 ± 2.02 a	10.62 ± 1.04 a	9.92 ± 1.19 a
6th instar	13.54 ± 3.23	—	—
7th instar	26.62 ± 3.66	—	—

Different letters denote a significant difference ( *p* < 0.05).

## Data Availability

Data will be made available on request.

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
