# Peer review of "Effects of Yeast on the Growth and Development of Drosophila melanogaster and Pardosa pseudoannulata (Araneae: Lycsidae) Through the Food Chain"

_insects, 2025, doi:10.3390/insects16080795_

Round 1
Reviewer 1 Report
Comments and Suggestions for Authors
This is a feeding study examining effects of yeast addition to the culture medium of Drosophila melanogaster on the fly itself, and on Pardosa pseudoannulata, a predator fed on the flies. Effects on the predator were weaker than on the fly, which is expected. To learn more from the study, it needs to be examined what changes yeast addition has – does it improve the supply of essential amino or fatty acids, or vitamins, to the fly? And what is known on the requirements with regards to those components for spiders and flies, respectively. The potential role of these micronutrients should be addressed in addition to the current focus on macronutrients, in order to improve our understanding of the foraging physiology and ecology of spiders.
L12: biological control of which insects in which crops? Why is it important to study the spider in the lab, then, since it seems unlikely that it is suitable for augmentation biological control. It would be good to make that connection clear.
L20 „shorter“ than what? Please revise this and the previous sentence to make clear which type of yeast had what effect on Drosophila.
L26: The statement that Drosophila glucose and total protein were affected by yeast contradicts the opposite statement in L24.
L26: Please re-structure: first everything about Drosophila, then everything about Pardosa, not back and forth.
L27: Is there no more to conclude? Nothing about Pardosa? I would say that the effects of medium on the prey are diluted towards the predator.
L32: Avoid valuing terms such as “excellent” – this is a question of perspective.
L36: Five references for this specific topic do not suggest that it is poorly studied. Has it been studied in wolf spiders before? Can you, from the beginning, also refer to the changes that yeast causes in the composition of the medium and of the fly, especially with respect to potential essential components? If no essential components are affected, this could explain why spiders show little response except for higher fattening.
L48: Can you please specify the length of the growth, number of egg laying periods etc.? This always depends on the context – some arctic wolf spiders grow for more than five years before being adult. Please also replace “yield” with “reproductive output”.
L65: Please add something on micronutrients, e.g. polyunsaturated fatty acids (e.g. https://doi.org/10.3389/fevo.2021.707570), and effects of yeast on them. Pardosa pseudoannulata can feed on Chironomidae, which are expected to have much higher PUFA content that herbivorous insects, which could be an important aspect of prey quality.
2.3: Please specify in what kind of containers the juvenile spiders were reared, and how many spiders per container.
L160: I would suggest to avoid abbreviations and write out “…time of D. melanogaster on activated yeast were significantly shorter than on inactive yeast (…”.
In the results, please try to describe the results in a more quantitative was. It is more informative to read “Protein content was 50% higher in Drosophila grown on both types of yeast than without yeast addition.” Than to just learn about significance levels. This is particularly interesting for the total fat content, which looks about doubled in AY relative to NY.
L267: yeast are not animals.
L295: The switch from physiology to ecology is very abrupt, in the middle of the paragraph. Please try to make a better transition. Clearly, nutrition of spiders is important also in the field, but I would be rather cautious: Is anything known on nutritional effects on fitness, populations or communities of spiders in the field? It should be clear if the implications for field populations are speculative or grounded on evidence, and potential research gaps identified.
L310: What exactly is “useful” about the information? What recommendations for spider rearing in the lab can you give? One important conclusion could be that a pure Drosophila diet, irrespective of yeast, does not allow Pardosa to complete its development. Pardosa breeding is a difficult task, and the literature should be screened for successful examples (please also check work by Ferenc Samu, https://www.researchgate.net/profile/Ferenc-Samu).
Author Response
This is a feeding study examining effects of yeast addition to the culture medium of Drosophila melanogaster on the fly itself, and on Pardosa pseudoannulata, a predator fed on the flies. Effects on the predator were weaker than on the fly, which is expected. To learn more from the study, it needs to be examined what changes yeast addition has – does it improve the supply of essential amino or fatty acids, or vitamins, to the fly? And what is known on the requirements with regards to those components for spiders and flies, respectively. The potential role of these micronutrients should be addressed in addition to the current focus on macronutrients, in order to improve our understanding of the foraging physiology and ecology of spiders.
Reply: We have revised the paper carefully, and we hope the current version will meet the standard of the journal.
L12: biological control of which insects in which crops? Why is it important to study the spider in the lab, then, since it seems unlikely that it is suitable for augmentation biological control. It would be good to make that connection clear.
Reply: We accept the suggestions. Pardosa pseudoannulata mainly preys on large and medium-sized pests such as moths and locusts. If the quantity can be expanded in the laboratory and then released back into the wild, the result of biological pest control can be achieved.
L20 „shorter“ than what? Please revise this and the previous sentence to make clear which type of yeast had what effect on Drosophila.
Reply: We accept the suggestions. We have revised these accordingly.
L26: The statement that Drosophila glucose and total protein were affected by yeast contradicts the opposite statement in L24.
Reply: We accept the suggestions. There are errors in the language description.
L26: Please re-structure: first everything about Drosophila, then everything about Pardosa, not back and forth.
Reply: We accept the suggestions. We have revised these accordingly in the Abstract.
L27: Is there no more to conclude? Nothing about Pardosa? I would say that the effects of medium on the prey are diluted towards the predator.
Reply: We accept the suggestions. The main content of our current experiment is the effect of adding different treatments of yeast to the fruit fly culture medium under laboratory conditions on the growth and development of the Pardosa pseudoannulata.
L32: Avoid valuing terms such as “excellent” – this is a question of perspective.
Reply: We accept the suggestions. We have revised these accordingly in the Introduction.
L36: Five references for this specific topic do not suggest that it is poorly studied. Has it been studied in wolf spiders before? Can you, from the beginning, also refer to the changes that yeast causes in the composition of the medium and of the fly, especially with respect to potential essential components? If no essential components are affected, this could explain why spiders show little response except for higher fattening.
Reply: We accept the suggestions. The five references cited are previous studies on the effects of other nutrients on spiders.
L48: Can you please specify the length of the growth, number of egg laying periods etc.? This always depends on the context – some arctic wolf spiders grow for more than five years before being adult. Please also replace “yield” with “reproductive output”.
Reply: We accept the suggestions. We have revised these accordingly in the Introduction. The growth length and the number of eggs laid by the ring-banded leopard spider vary with different temperatures and ages.
L65: Please add something on micronutrients, e.g. polyunsaturated fatty acids (e.g. https://doi.org/10.3389/fevo.2021.707570), and effects of yeast on them. Pardosa pseudoannulata can feed on Chironomidae, which are expected to have much higher PUFA content that herbivorous insects, which could be an important aspect of prey quality.
Reply: We accept the suggestions. We have revised these accordingly in the Introduction.
2.3: Please specify in what kind of containers the juvenile spiders were reared, and how many spiders per container.
Reply: We accept the suggestions. We have revised these accordingly.
L160: I would suggest to avoid abbreviations and write out “…time of D. melanogaster on activated yeast were significantly shorter than on inactive yeast (…”. In the results, please try to describe the results in a more quantitative was. It is more informative to read “Protein content was 50% higher in Drosophila grown on both types of yeast than without yeast addition.” Than to just learn about significance levels. This is particularly interesting for the total fat content, which looks about doubled in AY relative to NY.
Reply: We accept the suggestions. We have revised these accordingly.
L267: yeast are not animals.
L295: The switch from physiology to ecology is very abrupt, in the middle of the paragraph. Please try to make a better transition. Clearly, nutrition of spiders is important also in the field, but I would be rather cautious: Is anything known on nutritional effects on fitness, populations or communities of spiders in the field? It should be clear if the implications for field populations are speculative or grounded on evidence, and potential research gaps identified.
Reply: We accept the suggestions. We have revised these accordingly.
L310: What exactly is “useful” about the information? What recommendations for spider rearing in the lab can you give? One important conclusion could be that a pure Drosophila diet, irrespective of yeast, does not allow Pardosa to complete its development. Pardosa breeding is a difficult task, and the literature should be screened for successful examples (please also check work by Ferenc Samu, https://www.researchgate.net/profile/Ferenc-Samu).
Reply: We accept the suggestions. We have revised these accordingly.
Reviewer 2 Report
Comments and Suggestions for Authors
12: delete double colon on line
12: include authority and year of species and also idem in title.
16: Should be as a predator, not “its predator” as D. melanogaster is not part of the natural diet of this species.
27: the study significance is thus not about the spider, so why isn't the fruit fly the focus of the title/manuscript?
32: Add reference.
46: Never start sentence with abbreviated genus, write out the name
46: typo at beginning delete full stop after fields
86: see comment on line 16
91: bags to “sacs”
91: How were the spiders identified? Cite.
132: italicise
148: wrong font and size
275: here and throughout, the references are not in correct journal format
277: error with full stop
285: fifth instar, what about second instar? These were analysed according to previous statements
289: error with full stop
305: do not captilise species name
306: ibid
General: The real significance seems to be in results for the fruit flies and the spiders are a trivial matter, yet the manuscript leads with the spiders, I would reconsider this and switch around.
Author Response
12: delete double colon on line12:
12: include authority and year of species and also idem in title. 12:
Reply: We accept the suggestions. We have revised these accordingly.
16: Should be as a predator, not “its predator” as D. melanogaster is not part of the natural diet of this species.
Reply: We accept the suggestions. We have revised these accordingly.
27: the study significance is thus not about the spider, so why isn't the fruit fly the focus of the title/manuscript?
Reply: We accept the suggestions. We have revised these accordingly.
32: Add reference.
46: Never start sentence with abbreviated genus, write out the name
46: typo at beginning delete full stop after fields
86: see comment on line
91: bags to “sacs”
91: How were the spiders identified? Cite.
132: italicise
148: wrong font and size
275: here and throughout, the references are not in correct journal format
277: error with full stop
285: fifth instar, what about second instar? These were analysed according to previous statements
Reply: We accept the suggestions. The growth rate and age length of 2nd instar spiders respond more significantly to en-vironmental changes (such as temperature and food) than those of spiderlings (Jakob et al.,1996).
289: error with full stop
305: do not captilise species name
306: ibid
General: The real significance seems to be in results for the fruit flies and the spiders are a trivial matter, yet the manuscript leads with the spiders, I would reconsider this and switch around.
Reply: We accept the suggestions. We have revised these accordingly.
Reviewer 3 Report
Comments and Suggestions for Authors
In the work titled " Effects of nutrients on the growth and development of Pardosa 1 pseudoannulata (Araneae: Lycosidae) through the food chain” The authors analyze, using simple biochemical methods, the amounts of lipids, proteins, and carbohydrates found in the offspring of the fly D. melanogaster, after being fed three different cultures: with yeast, without yeast, and with inactivated yeast. The experiments are adequate, as are the data analyzed. My main problem is that, in my opinion, the title does not adequately reflect what the manuscript represents.
Possible alternative titles:
Effects of yeast on the growth and development of D. melanogaster and Pardosa pseudoannulata (Araneae: Lycosidae) through the food chain.
or
Effects of nutrients (yeast) on the growth and development of D. melanogaster and Pardosa pseudoannulata (Araneae: Lycosidae).
Regarding the discussion, the authors are correct when they state, "Our study shows the importance of yeast for the growth and development of fruit flies. Since a single food cannot provide sufficient nutrition for the spiders, we will conduct a mixed diet until the spiders mature in the follow-up experiment, and then study the effects of yeast on the spiders throughout the food chain. This study provides useful information on the nutritional requirements for the growth of D. melanogaster and predatory spiders." In that sense, the manuscript is a step forward in relation to the topic raised, but further studies are needed to fully understand the nutritional needs of spider development.
In the manuscript, the authors describe very well what happens with Drosophila, but they describe in little detail what happens in the spiders. There is no mention of vitellogenesis, vitellogenin, or yolk nutrients, which often remain internalized in the offspring.
The work has an analysis appropriate to what the authors propose, but if the title is not changed, they should add more information regarding the development and nutrition of spiders in general and P. pseudiannulata in particular.
Author Response
In the work titled " Effects of nutrients on the growth and development of Pardosa 1 pseudoannulata (Araneae: Lycosidae) through the food chain” The authors analyze, using simple biochemical methods, the amounts of lipids, proteins, and carbohydrates found in the offspring of the fly D. melanogaster, after being fed three different cultures: with yeast, without yeast, and with inactivated yeast. The experiments are adequate, as are the data analyzed. My main problem is that, in my opinion, the title does not adequately reflect what the manuscript represents.
Possible alternative titles:
Effects of yeast on the growth and development of D. melanogaster and Pardosa pseudoannulata (Araneae: Lycosidae) through the food chain. Or Effects of nutrients (yeast) on the growth and development of D. melanogaster and Pardosa pseudoannulata (Araneae: Lycosidae).
Regarding the discussion, the authors are correct when they state, "Our study shows the importance of yeast for the growth and development of fruit flies. Since a single food cannot provide sufficient nutrition for the spiders, we will conduct a mixed diet until the spiders mature in the follow-up experiment, and then study the effects of yeast on the spiders throughout the food chain. This study provides useful information on the nutritional requirements for the growth of D. melanogaster and predatory spiders." In that sense, the manuscript is a step forward in relation to the topic raised, but further studies are needed to fully understand the nutritional needs of spider development.
In the manuscript, the authors describe very well what happens with Drosophila, but they describe in little detail what happens in the spiders. There is no mention of vitellogenesis, vitellogenin, or yolk nutrients, which often remain internalized in the offspring.
The work has an analysis appropriate to what the authors propose, but if the title is not changed, they should add more information regarding the development and nutrition of spiders in general and P. pseudiannulata in particular.
Reply: We accept the suggestions. We have revised these accordingly.
Round 2
Reviewer 3 Report
Comments and Suggestions for Authors
I think the manuscript has now improved considerably.